# Recent Progress in the Vaccine Development Against Epstein–Barr Virus

**DOI:** 10.3390/v17070936

**Published:** 2025-06-30

**Authors:** Yihao Dai, Botian Zhang, Luming Yang, Shuo Tao, Yijing Yu, Conglei Li

**Affiliations:** 1School of Medicine, The Chinese University of Hong Kong, Shenzhen 518172, China; yihaodai@link.cuhk.edu.cn (Y.D.); botianzhang@link.cuhk.edu.cn (B.Z.); lumingyang@link.cuhk.edu.cn (L.Y.); 122090507@link.cuhk.edu.cn (S.T.); yuyijing@cuhk.edu.cn (Y.Y.); 2Guangdong Basic Research Center of Excellence for Aggregate Science, School of Science and Engineering, The Chinese University of Hong Kong, Shenzhen 518172, China

**Keywords:** Epstein–Barr virus, vaccine, glycoprotein

## Abstract

The Epstein–Barr virus (EBV) is the first human herpesvirus identified as an oncogenic agent, with approximately 95% of adults worldwide being latently infected. EBV infection is associated with multiple diseases, including nasopharyngeal carcinoma, Hodgkin’s lymphoma, infectious mononucleosis, and multiple sclerosis. Given significant EBV-associated disease burden, developing effective vaccines against EBV remains a priority. In this review, we first presented the current understanding of EBV biology and pathogenesis, focusing on its biological structure and immune evasion mechanisms, and discussed key viral antigens—including gp350, gp42, gH/gL, and latency proteins—as potential targets for EBV vaccine development. We also summarized recent advances in various EBV vaccine platforms, including subunit, viral vector-based, nanoparticle-based, and mRNA vaccines, and discussed the related preclinical and clinical evidence, although no effective EBV vaccine has been approved for clinical use yet. In summary, this review provides an overview of the current landscape in EBV vaccine research, and sheds new light on developing new therapeutic approaches against EBV-associated diseases.

## 1. Introduction

The Epstein–Barr virus (EBV) was the first herpes virus confirmed to be associated with human cancers in 1964. Currently, it is estimated that approximately 95% of adults worldwide are latent carriers of this virus [1]. EBV infection has been etiologically linked to a spectrum of clinical manifestations, ranging from benign conditions (e.g., infectious mononucleosis) to severe autoimmune disorders (including systemic lupus erythematosus, multiple sclerosis, and rheumatoid arthritis), lymphoproliferative malignancies (notably Burkitt’s lymphoma and Hodgkin’s lymphoma), and epithelial carcinomas (particularly nasopharyngeal carcinoma [NPC]) [2,3,4]. EBV infection is associated with approximately 1% of all human cancer cases, and accounts for 2% of all cancer-related deaths. Every year, there are over 200,000 EBV-related cancer cases and 140,000 deaths worldwide [5].

NPC is prevalent in east and southeast Asia. Currently, radiotherapy and chemotherapy remain the primary treatment modalities for EBV-associated NPC patients. However, NPC recurrence and metastasis are frequently encountered challenges and are strongly linked to unfavorable outcomes and poor prognosis [6,7]. As early as 1976, Dr. M.A. Epstein, the discoverer of EBV, proposed the development of a preventive vaccine against EBV as a strategy to prevent EBV infection and to reduce the burden of EBV related cancers [8]. Since then, for over 40 years, prophylactic EBV vaccines aimed at preventing primary EBV infection have been under intensive development. From the 1970s onwards, multiple EBV vaccine studies have encompassed subunit vaccines, epitope vaccines, DNA vaccines, nanoparticle-based vaccines, viral vector vaccines, virus-like-particles (VLPs), or dendritic cell (DC) vaccines [9]. So far, although many vaccines have completed clinical trials, none of them has been able to effectively prevent EBV infection in humans [10]. Due to the fact that EBV itself contains multiple antigens that are highly carcinogenic, and it is difficult to amplify EBV through in vitro cultivation, it is not possible to obtain inactivated or attenuated vaccines against EBV [9].

In this review, we summarize the features of the biology and immune evasion of EBV, the key target antigens for EBV vaccines, recent advances in various EBV vaccine platforms, and the related preclinical and clinical evidence for evaluating these vaccines.

## 2. The Biology and Immune Evasion Mechanisms of EBV

EBV exhibits significant dormancy ability in most individuals after infection, and directly causes various symptoms only in a minority of affected individuals [11]. After EBV infection, the incubation period is usually prolonged, and in some cases, no symptoms may ever appear, which makes it difficult for the human immune system to recognize and clear EBV in a timely manner [12]. It is crucial to understand the basic biology of EBV, including its viral structural components, life cycle, and host invasion strategies, in order to accurately analyze the immune escape mechanisms and design targeted vaccines against EBV infections.

### 2.1. EBV Biology

EBV is one of the eight known human herpesviruses [13]. It has a linear double-stranded DNA genome, approximately 170 kb in length, encoding 85 proteins and 46 functional small untranslated RNAs [14]. Some proteins are involved in replicating the viral genome and producing new viral particles in the lytic cycle (viral production). Herpesviruses also have a second gene expression pattern, allowing them to persist in infected hosts for a long time without producing viral particles, and to express different viral genes during this latent period [15]. There is a polyhedral protein capsid around the core of the EBV genome, consisting of 162 capsid units [16]. The outermost layer is a lipid envelope with surface glycoprotein protrusions. The EBV genome encodes nine unique envelope glycoproteins, which provide the structural basis for EBV invasion into host cells [17]. At the same time, these glycoproteins have high antigenicity due to exposure, and they are important targets for EBV-specific vaccines.

EBV transmission is mainly through saliva. However, body fluids like breast milk, and organ transplants can also transmit the virus from one host to another [18]. EBV viral particles target the epithelial cells and B cells of the oropharynx when entering the new host. Infection is often asymptomatic in early childhood but causes infectious mononucleosis (IM) when acquired in adolescence or adulthood [12]. In addition, EBV is also associated with several autoimmune diseases. The EBV nuclear antigen 1 (EBNA-1) can induce the production of antibodies reactive to double-stranded DNA (dsDNA) in humans, thereby triggering systemic lupus erythematosus (SLE) [19]. EBV infection and its subsequent effects are a multistep process. Initiated by exposure to the oropharynx, where cell infection occurs, the process ultimately culminates in the lifelong persistence of EBV within circulating memory B cells, effectively accompanying the host throughout their lifetime [20].

The mechanisms by which EBV enters host cells share many similarities with other herpesviruses [17,21]. When exposed to susceptible areas of the human body (usually the respiratory tract), the virus attaches to the surface of cells and then infects through membrane fusion. This attachment is mediated by various viral glycoproteins and binding receptors. The gH/gL heterodimer and the viral fusion protein gB are essential for viral entry across all herpesviruses [22]. Unlike other herpesviruses, EBV has two different invasion mechanisms that infect the epithelial cells and B cells, respectively. On the surface of epithelial cells, gH/gL directly binds to the ephrin receptor tyrosine kinase A2 (EphA2) to trigger gB insertion into the host cells [23] (Figure 1). Previously, Zhang et al. identified EphA2 as a key participant in the EBV epithelial cell entry through CRISPR-Cas9 knockout of EphA2, overexpression of EphA2, and use of EphA2 inhibitors [24]. For B cell infection, the binding of major viral glycoprotein 350 (gp350) to CD21 (CR2) is the main viral receptor, while additional involvement of gp42 is required to form a heterotrimer with gH/gL, which is then recognized by human leukocyte antigen (HLA) class II molecules on the surface of the B cells and triggers endocytosis [21,25,26].

### 2.2. EBV Immune Evasion Mechanisms

Due to the high cell turnover rate of the epithelial cells, latent EBV is more commonly found in memory B cells, which can live for months or years in vivo. The lifecycle of EBV includes the following three stages: infecting host cells, delaying latency, and lytic activation [27]. The reason why EBV is difficult to identify and eliminate is that it may be accompanied by lifelong latency. During latency, EBV only expresses a small amount of proteins, and these proteins are not released outside the host cell or presented on the cell surface, so they cannot be detected by the host immune system, thus achieving immune evasion [28]. For the detailed immune evasion mechanisms EBV adopts for its benefits, please refer to our previous review paper [3].

## 3. Key EBV Antigens as the Vaccine Candidates

Antigen selection is a critical step in the design of the EBV vaccine, as it determines the vaccine’s ability to elicit robust and protective immune responses. As mentioned above, EBV encodes 85 kinds of proteins and 46 functional small untranslated RNAs [14]. EBV glycoproteins, lytic proteins, and latent proteins are potential immunogens in the EBV vaccine design, especially those utilized by EBV for its entry into host cells (Figure 1). Here, we have compiled some EBV antigens that have been studied as vaccine targets and potential target antigens that may serve as breakthroughs in future vaccine research.

### 3.1. gp350

The gp350 in EBV is the most popular antigen target for EBV vaccine development, as it is the most abundant glycoprotein on the envelope of EBV, and has the highest immunogenicity. Furthermore, gp350-mediated EBV infection leads to lifelong infection of the B cell [26]. EBV gp350 uses a negatively charged concave surface to bind to CR2 on the B cell surface, simulating the interaction between complement C3d and CR2 at the amino acid residue level [29]. EBV gp350 can be detected on the surface of host cells throughout the entire lifecycle of EBV, including latency [30,31]. It is reported that gp350 can be detected in EBV positive NPC cell lines and primary EBV-related cancer samples from patients [32,33].

As mentioned above, gp350 has been a popular preventive subunit vaccine development target [4]. Jackman et al. first produced a gp350 subunit vaccine candidate free of gp220 isoform via modified Chinese hamster ovary (CHO) cells in 1999, which caused immune reactions with high neutralizing antibody titers in rabbits [34]. Those antibodies were able to neutralize wild-type EBV, demonstrating the potentiality of gp350 as a subunit vaccine for EBV. The gp350 appears as monomeric or multimeric proteins in different preclinical trials with or without adjuvants [10]. The gp350 candidate vaccine protects the B cells, but cannot protect the epithelial cells from EBV infection.

### 3.2. gp42

The EBV gp42 is a type II membrane protein. It is essential for EBV entry into the B cells but is not required for epithelial cell infection [35] (Figure 1). EBV has a strict cellular tropism, which refers to the virus’s ability to selectively infect specific host cell types. Tropism switching, on the other hand, denotes the process by which EBV alters its recognition and infection preference for host cell types during the infection course, through regulating gene expression and viral protein functions. The relative amounts of two-part gH/gL and three-part gH/gL-gp42 complexes in EBV virions impact tropism: epithelial cell-derived viruses (rich in gp42) are more infectious for the B cells, while the B cell-derived viruses (rich in gH/gL) favor epithelial infection [36]. This supports a model where the virus from latently infected B cells tends to infect the epithelial cells, and the virus from the epithelial cells prefers to target the B cells [37]. As an orally transmitted virus, saliva-shed EBV mainly derives from HLA class II-negative cells, with transmission possibly via direct oral B-cell infection or epithelial infection (via gp350/220) before B-cell access. Therefore, gp42 regulates the tendency of EBV to infect different host cell types. In addition, gp42 may help the virus evade the immune system: gp42 binds to HLA class II molecules and inhibits the presentation of HLA class II restricted antigens to T cells [38].

Bu et al. identified two distinct fragile sites for receptor binding and B cell fusion on gp42, resulting in the isolation of two gp42 monoclonal antibodies (mAbs) [39]. Among them, mAb A10 was passively transferred to humanized mice and showed nearly 100% protective effect against viremia and EBV lymphoma after EBV attack. Furthermore, Kong et al. obtained blood samples from 129 NPC patients and 387 individual as matched control in southern China, and evaluated the effect of anti-gp42 IgG on NPC using ELISA assays. The gp42-specific IgG titers were significantly reduced in NPC patients, and the protective effects of gp42 IgG were observed in NPC patients diagnosed at 5 years or more, 1–5 years, and less than 1 year after blood collection. The gp42-specific IgG titers increased, while the malignancy of the NPC decreased. Elevated IgG titers against EBV gp42 can reduce the risk of NPC, indicating that gp42 is a potential EBV prophylactic vaccine target [40].

### 3.3. gH/gL

The gH and gL proteins are also surface glycoproteins of EBV, with molecular weights of approximately 90 kDa and 25 kDa, containing 706 and 137 amino acids, respectively. Both of them are conserved components of all herpesviruses [41]. The gL protein is inserted between the gH domains H1A and H1B, and non-covalently binds to gH in a 1:1 ratio to form a conserved gH/gL heterodimer complex, which is an important structure for the herpes virus to enter host cells [42,43] (Figure 1).

Anti-gH/gL neutralizing antibodies can prevent viral entry by interfering with the fusion process between the virus and the host cell membrane. As a more broad-spectrum antigen, compared with the gp350 and gp42 antibodies, the gH/gL antibody can not only block EBV infection of the epithelial cells, but also is expected to be effective against other herpes viruses [44]. It has been reported that anti-gH/gL antibodies are the main determinant of EBV neutralization in human plasma. Researchers have used a human endogenous immunoglobulin depletion assay to compare the neutralization effects of different glycoprotein antibodies, and antibodies against gp350, gH/gL, and gp42 contributed 44.6% ± 4.37%, 46.9% ± 3.29%, and 10.9% ± 1.85% of B cell neutralization, respectively, while gH/gL antibodies contributed to 76% ± 0.89% of epithelial cell neutralization [45].

### 3.4. Latency Proteins

In addition, EBV also expresses latent proteins that do not participate in virus entry, such as LMP1 and LMP2 expressed only in the latent phase II, as well as EBNA1 expressed in all three latent phases [4]. LMP1 and LMP2 are two different latent membrane proteins, and participate in various cellular pathways, which are related to tumor development and anti-apoptotic function [46,47,48]. EBNA1 is a potent immunogen that is the only protein found in all EBV related cancers [49,50]. It usually exists in the form of homodimers and is crucial for maintaining the viral genome and continuous replication of the virus [51].

These proteins are almost exclusively expressed within host cells, and their expression levels are highly conserved during the latent period. Conventional immune methods, including corresponding vaccines and monoclonal antibodies, are difficult to target these latency proteins due to their silent expression within host cells. Although studies have shown that the EBNA1 specific memory B cells can be detected in patients, it is still difficult to target the EBV infected cells during the latent period [52,53].

## 4. Development of Preventive EBV Vaccines

As mentioned above, EBV infection is related to various diseases, including nasopharyngeal carcinoma, Hodgkin’s lymphoma, infectious mononucleosis, and multiple sclerosis [54]. Currently, no preventive or therapeutic EBV vaccine is available in the market (Figure 2). As Zhong et al. mentioned, it is necessary and urgent to develop prophylactic vaccines to prevent EBV infection and to reduce the burden of EBV-related diseases [9]. However, the oncogenicity effects of specific EBV genes and the limited production of viral particles in vitro have restricted the development of inactivated or attenuated EBV vaccines [9,55].

### 4.1. Adjuvants of EBV Vaccines

Adjuvants are used to enhance and prolong the effects of the vaccines [56]. The functions of adjuvants are related to many mechanisms. First of all, it can boost the immune responses elicited by vaccines [57]. Some vaccines made of recombinant proteins or peptides have low immunogenicity [58], but adjuvants can overcome this shortcoming [56]. Secondly, adjuvants can modulate immune polarization [59], and regulate different immune cells to function. For example, Th1 cells fight with intracellular pathogens which can be regulated by monophosphoryl lipid A (MPL) adjuvant [60]. The function of Th2 cells, including clearing up extracellular pathogens and promoting allergies [61], can be boosted by Alum adjuvant [62]. Cytosine–phosphate–guanine (CpG) and AS01 adjuvants can activate cytotoxic T cells for the vaccines [63]. Adjuvant effects also vary depending on the delivery routes [57]. Alum and some oil-based emulsions can form a depot which will slower the release of the drug and delay the clearance [62]. It can not only prolong the therapeutic effect, but also weaken the side effects. Nanoparticles can directly target the antigens to dendritic cells, and make the immune response faster [64]. In addition, adjuvants (e.g., MPL and CpG) can trigger pattern recognition receptors to activate innate immune response [58,65].

There are different types of adjuvants used in EBV vaccines: mineral salts, oil-based emulsions, nanoparticles, bacterial derivatives, and some new material adjuvants [10]. In mineral salts, alum is the most common adjuvants [62]. It is always mixed with other adjuvants to increase the efficacy of the vaccines, including lipid A [61] and CpG oligodeoxynucleotide (CpG-ODN) [66]. Freund’s adjuvant [67] (both complete and incomplete) and the Sigma adjuvant system (50% *v*/*v*) (SAS) are oil-based emulsions [68]. The SAS is made of light mineral oil (Drakeol) and an emulsifier (Arlacel A) [69]. It has mild innate immune stimulation and balanced Th1/Th2 effects [70]. But it is more widely used in veterinary vaccines. Nanoparticles (NPs) are more advanced adjuvants than mineral salts and oil-based emulsions [71]. They can boost comprehensive immune response in Th1/Th2 cells, CD8^+^ T cells and can also assist to produce higher affinity and long-lasting antibody [72]. NPs can be vectors for some fragile antigens (DNA and RNA), and it is suitable for mucosal immune responses, which can be applied in oral and nasal vaccines [73]. Apart from that, NPs have fewer side effects than other adjuvants and can be degraded by the body [74]. *H. pylori* ferritin-based NPs are the most common NPs used in EBV vaccines [75]. In addition, TiterMax®, immune stimulating complexes (ISCOM), Syntex adjuvant formulation-1 (SAF-1) and *Bordetella pertussis* bacterial cells are also very widely used as adjuvants in EBV vaccines formulation [10].

### 4.2. Subunit Vaccines

A double-blinded, randomized clinical phase I/II trial was performed in Belgium and Brussels using the gp350 subunit vaccine. The vaccine was produced by GlaxoSmithKline Biologicals (GSK, Rixensart, Belgium), containing 50 µg of gp350, either non-adjuvanted or adjuvanted with aluminum salt (Alum) or aluminum salt plus 3-O-desacyl-4′-monophosphoryl lipid A (AS04) [76]. Both the safety and immunogenicity of this gp350 subunit vaccine were evaluated in this trial. The induction of both humoral and cellular immunity was proved. Every participant generated anti-gp350 antibodies [76]. However, not every participant generated EBV-neutralizing monoclonal antibody 72A1, which has a promising role in defending against EBV infection in the B cells in vitro [77]. Nonetheless, one serious adverse event out of a total of 67 participants in phase I was reported, possibly related to the vaccination [76].

Another phase II, double-blinded, randomized, placebo-controlled clinical trial was conducted in healthy young adults, demonstrating the similar immunogenicity of a gp350 subunit vaccine, which was formulated by GlaxoSmithKline Biologicals. Each dose of the vaccine contained 50 µg of gp350 with the AS04 adjuvant system containing aluminum hydroxide and 3-O-desacyl-4′-monophosphoryl lipid A [78]. The results demonstrated that the vaccine exhibited excellent profiles of safety and immunogenicity, with 98.7% of the subjects seroconversion to the anti-gp350 antibody (95% CI: 85.5–97.9%) at 1 month after the third vaccination, and the antibody persisted for more than 18 months. A total of 69.86% (95% CI, 58.00–80.06%) of the subjects showed seroconversion at 6 months after the first injection. More importantly, the vaccine showed significant efficacy against infectious mononucleosis caused by the EBV infection, with a protective efficacy of 78% (95% CI: 1.0–96.0%), but it did not prevent the development of asymptomatic EBV infection [78].

In addition to healthy individuals, the safety and immunogenicity of a gp350 subunit vaccine were also evaluated in immunosuppressive patients: EBV-negative children awaiting kidney transplantation (phase I trial). The gp350/0.2% alhydrogel vaccine was produced by the Cancer Research UK Formulation Unit, similar to Jackman et al.’s protocol described previously [34]. A total of 16 children awaiting kidney transplantation were divided into two groups (6 or 10 patients per group), with vaccine doses of 12.5 µg and 25 µg, respectively, for either three or four cycles. Safety was proven with only two systemic reactions (25 µg group, fever) and eight mild injection site reactions (12.5 µg group, 1/6; 25 µg group, 5/10) reported. One participant developed EBV-related post-transplant lymphoproliferative disorders (PTLD). After excluding the participants with EBV seroconversion that occurred at undefined time points during follow-up, all 13 evaluable participants developed anti-gp350 antibody responses (median peak levels: 607 units for 12.5 µg group; 612 units for 25 µg group). However, only four participants developed neutralizing antibodies (1/4 in 12.5 µg cohort; 3/9 in 25 µg cohort). Four asymptomatic EBV infections occurred during active follow-up. The post-transplant EBV load was similar to that of nonvaccinees during the first 26 weeks post-transplantation (8 vaccinees, 31 nonvaccinees). It indicated the gp350 subunit vaccine did not affect the post-transplantation EBV load. Due to the lack of a control group, the efficacy of this vaccine cannot be evaluated [79]. The immunosuppressive status of those participants could be responsible for the suboptimal performance of this vaccine.

In addition to neutralizing antibodies, a new approach to generate the subunit EBV vaccine is activating EBV-specific CD8^+^ T cells to control the EBV-infected B cells. Elliott et al. (2008) conducted a phase I trial with the HLA B*0801-restricted CD8 T-cell epitope FLRGRAYGL (FLR) from EBNA3, formulated with the adjuvant tetanus toxoid (TT) and emulsified with the water-in-oil adjuvant Montanide ISA 720 [80]. This vaccine was safe with only mild-to-moderate injection-site reactions (e.g., swelling, erythema) and no serious adverse events. FLR-specific T cells were detected in eight of nine vaccinees but in zero of four placebo recipients. The T cell receptor (TCR) sequences of those FLR-specific T cells elicited by the vaccination matched the naturally produced EBV-specific T cells, confirming the physiological relevance. All four vaccinees who acquired EBV post-vaccination remained asymptomatic. One of two placebo recipients developed infectious mononucleosis, while no vaccinees showed the disease. Interestingly, seroconverted vaccinees developed normal CD8^+^ T responses to the other EBV epitopes of EBNA3, including QAKWRLQTL (QAK) and RAKFKQLL (RAK), without any skewing in immunodominance, which revealed the broad immunity generated by this vaccine [80].

### 4.3. Viral Vector-Based EBV Vaccines

Michael Mackett and John R. Arrand first used the vaccinia virus as a platform for EBV vaccines [81]. They inserted the isolated gp340 (a historical name of gp350) gene into vaccinia virus strains WR (laboratory) and Wyeth (vaccine) under a vaccinia promoter. In vitro experiments confirmed the expression of the gp340 proteins. Two rabbits were vaccinated with the recombinant WR-gp340 vaccinia virus. High titers of anti-gp340 and neutralizing antibodies were detected. The results obtained support the use of vaccinia-based EBV vaccines and the immunogenicity of the gp350 proteins [81].

Mackett et al. utilized a recombinant vaccinia virus, vMA1, which expresses the EBV gp340, to test immunogenicity in marmosets. Low titers of anti-gp340 antibodies in two of four animals immunized with vMA1 were detected after the first dose, and all four marmosets developed anti-gp340 antibody titers after the second vaccination. Lower whole mouth fluid (WMF) EBV DNA detection rate after the EBV challenge was revealed in the vaccinated group (42%, 5/12) compared with controls (67%, 14/21) [82].

Similar efforts were made previously by Morgan et al. in 1988. Two different vaccinia virus strains, the WR laboratory strain and the Wyeth vaccine strain (New York City Board of Health), were tested for their potential to be viral vectors for EBV vaccines. Twelve cottontop tamarins were assigned to three equal groups: control (no vaccine), recombinant WR-gp340 vaccine, and recombinant Wyeth-gp340 vaccine groups. High-dose EBV challenges were performed 6 weeks post-immunization. Anti-gp340 antibodies were neither detected in any of the vaccinated animals nor in the control group. Very low titers of the neutralizing antibodies could be detected in the WR-gp340 group. However, in terms of protecting EBV-induced malignant lymphomas, three of four animals in the WR-gp340 strain recombinant group showed complete protection (no tumors), and the remaining one exhibited delayed tumor progression. By contrast, no protection effect was observed in the Wyeth-gp340 or control group, except for one spontaneous tumor regression in the Wyeth-gp340 group [83]. These results demonstrated that the WR laboratory strain of the vaccinia virus is a better viral vector for the EBV vaccine.

The first human trial of the viral vector-based EBV vaccine was conducted in Beijing, China. In this phase I trial, Gu et al. utilized the licensed vaccinia strain, Tien Tan, to produce a live recombinant virus expressing membrane antigen BNLF-1 MA (gp220–gp340). This trial involved three different human populations: adults, juveniles, and infants. Safety was confirmed, with mild adverse effects shown. However, immunogenicity varied among the different populations. There were no significant differences in anti-gp350 antibodies before and after immunizations in the adult group. In the juvenile and infant groups, an increase in EBV-neutralizing antibodies was detected. Anti-vaccinia and anti-gp350 antibodies were also detected in the majority of participants in these two groups. It is worth noting that all 10 infants in the control group developed EBV infection during the 16-month follow-up period via natural routes, while 3 of 9 vaccinated infants developed EBV infection. It might suggest that the EBV vaccine effectively reduced the infection but did not prevent it. The negative data from the adult group might be due to previous exposure to the vaccinia virus [84]. However, no further experiments based on this study have been published. The possible reasons might be the potential adverse events of the live vaccinia virus [85].

Lockey et al. created a mixed vaccine comprising four types of vaccinia virus (VV) vectors, each expressing different EBV antigens: two lytic proteins (gp350 and gp110) and two latent proteins (EBNA-2 and EBNA-3C). Antibodies against the respective EBV proteins were detected following immunization of C57BL/6 female mice with each individual viral vector and the mixed vaccine formulation. Low titers of neutralizing antibodies against gp350 were detected in the VV-expressing gp350 only group and the mixed vaccine group, respectively. CD4^+^ T-cell responses to EBNA-2, a representative of the latent EBV antigens, were observed in both the VV-expressing EBNA-2 only and the mixed-vaccine groups, respectively [86].

Zhang et al. selected three peptides from the gp350 receptor-binding domains (RBD) and inserted them into a chimeric VLP formed by the C-terminally truncated Hepatitis B virus core (HBc) protein [87]. They tested different tandem order combinations of those three peptides and found that only two configurations (named 149-3A and 149-3B) elicited high antibody titers against gp350ECD123 (corresponding to gp3501-425) while other configurations generated low humoral responses. That might suggest that conformational changes, such as folding and flexibility, affect the immunogenicity of a peptide-based vaccine. This peptide-based virus-like particle vaccine elicited higher anti-gp350 antibody titers than the monomeric gp350ECD123 protein. A competition ELISA against mAb 72A1 was performed to demonstrate the neutralizing antibodies induced by this vaccine. The authors found a higher neutralizing antibody response in 149-3A and 149-3B vaccinated mice compared to monomeric gp350ECD123 immunized mice. An in vitro neutralization assay showed stronger neutralizing efficiency (by blocking EBV infection of AKATA cells) in 149-3A and 149-3B immunized mice compared to monomeric gp350ECD123 immunized mice.

### 4.4. Nanoparticle Vaccines

Sun et al. designed a nanoparticle (NP) vaccine, gB-I53-50 NP, by using the EBV fusion protein glycoprotein B (gB) [88]. They chose a de novo-designed icosahedral nanoparticle scaffold I53-50A1 as the backbone to present multiple copies of the gB proteins, and confirmed superior structural stability under thermal or chemical stress and non-inferior antigenicity, when compared to the subunit gB protein alone. Experiments in BALB/c mice showed that increasing copies of gB in the gB-NP induced higher anti-gB antibody titers. Thus, gB-NP with 100% gB valency was chosen for further evaluation.

The concomitant use of adjuvant MF59 further enhanced the immunogenicity of gB-NP. A direct comparison of the antibody titers between the gB-NP and gB groups showed that the highest gB-NP/gB binding IgG ratio was 14.5-fold in the MF59 adjuvant group and 17.7-fold in the unadjuvanted group at week 20 post-immunization. The addition of the MF59 adjuvant enhanced immunogenicity in both groups, but exhibited differential effects. While it increased anti-gB IgG titers in both the gB-NP and gB groups, the enhancement was more pronounced in the gB group. Consequently, the gB-NP/gB IgG ratio at week 20 in the adjuvant group was lower than in the unadjuvanted group, reflecting the adjuvant’s greater boosting effect on the non-nanoparticle formulation. Also, gB-NP was consistently superior in eliciting neutralizing antibodies compared to gB. However, the MF59 adjuvant can somehow compensate for gB in eliciting neutralizing antibodies. Anti-gB IgG titers and sera-neutralizing titers elicited by gB-NP were significantly higher than those elicited by gB. Moreover, the superior antibody titers elicited by gB-NP further improved as the observation time prolonged. Regarding cellular immune responses, when compared to the gB alone group, gB-NP exhibited enhanced performance in inducing immune-activating cytokines (IL-2, TNF-α, and IL-4), particularly among CD4^+^ T cells, further underscoring its immunostimulatory advantage. Experiments in non-human primates (NHP), *Macaca fascicularis*, revealed similar results as mice immunization. The NHP experiments further proved the safety of gB-NP, as no apparent adverse effects were observed. Interestingly, when polyclonal antibodies isolated from mixed sera of all vaccinated NHPs were injected into mice, followed by the EBV challenge, gB-NP showed significant superiority in preventing the lethal challenge of EBV and weight loss. The EBV DNA load also increased slowly, and reached a lower level in the gB-NP group, when compared to the gB group. Subsequent anatomical analysis indicated lower EBV DNA load in different organs and better histological appearance (less dysplastic cell proliferation) in the gB-NP group, indicative of protection from EBV-associated lymphoma [88].

Kanekiyo et al. analyzed different gp350 truncation variants [89]. They found that the smallest variant (D12) is unstable, and the largest variant (ectodomain) cannot be appropriately expressed in ferritin and encapsulin platforms. Finally, they selected a truncated variant of gp350, named D123 (425aa, contains three different binding domains), which can be stably expressed in the ferritin and encapsulin platforms, as the immunogen presented by self-assembled nanoparticle ferritin or encapsulin scaffolds. This nanoparticle maintained the native conformation of the gp350 ectodomain and preserved its immunogenicity, which was proved by a monoclonal antibody binding experiment using mAb 72A1 (targeting the CR2-binding site (CR2BS) on gp350) and mAb 2L10 (not competing with 72A1). Both D123-ferritin and D123-encapsulin induced significantly higher titers of the gp350-specific antibodies than the gp350 ectodomain alone (around 100-fold). Even a 10-fold lower dose (0.5 μg) of these two nanoparticles could elicit similar antibody titers, indicating the dose-sparing effects while maintaining high immunogenicity. NHP experiments were performed in *Macaca fascicularis*. Both nanoparticles increased the preexisting neutralizing antibody titers (caused by the natural infection with a lymphocryptovirus) by 119-fold (D123-ferritin) and 25-fold (D123-encapsulin) from the baseline after a third dose immunization, 3–10 times higher than that generated by the gp350 ectodomain alone. Challenge experiments with the vaccinia virus expressing EBV gp350 revealed that 80% (four of five) D123-ferritin-immunized mice were protected from severe weight loss, while almost all mice immunized with the gp350 ectodomain and the control mice lost more than 30% of their weight within 8 days post-infection. However, no obvious protection was observed in the D123-encapsulin group. The antibody specificity towards the CR2-binding site (CR2BS) was higher in the gp350-nanoparticles group than the gp350 ectodomain group, indicating that nanoparticle display enhances immunodominance of CR2BS, the critical site for potent virus neutralization [89].

Bu et al. found that anti-gH/gL antibodies are the principal neutralizing components against EBV infection in the epithelial cells (~75% of total activity), while anti-gH/gL antibodies only contribute 15–20% of the total neutralizing activity against EBV in the B cells [45]. They designed two self-assembled nanoparticles, expressing either gH/gL or gH/gL/gp42. Each epitope was intact and accessible on the nanoparticles, which was proved by neutralizing monoclonal antibody binding. The spatial conformation of these epitopes was optimal for cross-linking B cell receptors. Mice were immunized with 0.5 μg of soluble gH/gL, soluble gH/gL/gp42, gH/gL-ferritin, or gH/gL/gp42-ferritin formulated with the Sigma Adjuvant System (SAS) adjuvant. Nanoparticles displayed significantly enhanced immunogenicity compared to the soluble antigens. Mice immunized with gH/gL-ferritin or gH/gL/gp42-ferritin (both adjuvanted with SAS) elicited significantly higher binding antibody titers against gH/gL and gp42 than their soluble counterparts. Ferritin nanoparticle vaccines demonstrated significantly stronger neutralization than soluble protein vaccines in both the B cells and the epithelial cells. In the B cells, gH/gL-ferritin induced ~400-fold higher IC50 titers than soluble gH/gL, while gH/gL/gp42-ferritin showed about 200-fold improvement. The advantage was even more pronounced in the epithelial cells, with 3000-fold and 600-fold higher neutralization for gH/gL-ferritin and gH/gL/gp42-ferritin, respectively. Nanoparticle vaccines sustained high neutralizing titers (>3 months) without boosting, outperforming soluble proteins in both the B and epithelial cells. The gp42 selectively enhanced the B cell (but not epithelial) responses, highlighting antigen-specific design needs. The authors also evaluated combinations with gp350-ferritin. The results showed higher EBV B cell neutralizing titers in sera from mice immunized with gp350-ferritin plus gH/gL-ferritin or gp350-ferritin plus gH/gL/gp42-ferritin than any single regimen of these three vectors. However, the epithelial cells did not reveal the superiority of a combination regimen over a single regimen, potentially due to the overexpression of CR2 in the target cells. Similar experiments were performed in cynomolgus macaques (*Macaca fascicularis*). The results supported the findings from mouse experiments. The protection mechanism of those two nanoparticle vaccines might be due to the antibodies generated by them potently inhibiting virus glycoprotein-mediated B cell and epithelial cell membrane fusion, and gH/gL/gp42-ferritin was more potent in inducing B cell fusion inhibitory antibodies than gH/gL-ferritin. Antibodies induced in response to nanoparticle vaccines presenting either gH/gL-ferritin or gH/gL/gp42-ferritin specifically bound to the viral fusion domain of gH/gL, targeting identical epitopes to those identified by the reference monoclonal antibodies CL40, 769B10, and AMMO1 [45].

In addition to a single-type nanoparticle-based EBV vaccine, Zhong et al. recently developed a cocktail vaccine containing three types of nanovaccines targeting EBV antigens gH/gL, gB, and gp42, respectively. They used DOTAP, poly (lactic-co-glycolic) acid (PLGA), and 1,2-dithioaryl-sn-glycero-3-phosphoethanolamine-N-[(polyethylene glycol)-2000] (DSPE-PEG2000) to co-deliver adjuvants CpG and MPLA with three kinds of EBV antigens. The murine immunization experiments have shown that these cocktail nanovaccines can be internalized by antigen-presenting cells and transported to local draining lymph nodes, and also enhance germinal center formation, thus inducing effective humoral and cellular immune responses without significant side effects. The encapsulation of nanoparticles and the co-delivery with adjuvants significantly enhanced the immunogenicity of the EBV gH/gL, gB, and gp42 antigens. The anti-EBV IgG protective effects produced by the cocktail nanovaccine were stronger than those of the corresponding single antigen nanovaccine [90].

Recently, Li et al. generated a ferritin-based nanoparticle vaccine based on epitope peptides from the gp350 RBD [91]. They selected four peptides in RBD and one peptide from another domain of gp350 as the antigen. These five peptides were concatenated by a flexible linker (GGGGS) to construct a recombinant protein named L350, and L350 was anchored to the ferritin nanoparticle surface through irreversible binding of the SPY tag (C-terminus of L350) and the Catcher protein (N-terminus of the ferritin protein), which were cloned onto the respective sites. In vivo experiments demonstrated the potent humoral response elicited by the L350-ferritin nanoparticle vaccine. It is notable that the L350-ferritin nanoparticle, L350 protein, and gp350D123 protein (a commonly used truncated gp350 antigen) all were able to induce high titers of both the gp350D123 protein-specific antibody and L350 protein-specific antibody, which indicated the full accessibility of those epitopes both in recombinant nanoparticles and gp350 proteins. However, the L350-ferritin nanoparticle vaccine elicited the highest titers of both antibodies at peak value at week 10 among all the injected candidates (i.e., PBS, ferritin, L350-ferritin nanoparticle, L350 protein, and gp350D123 protein). The L350-ferritin nanoparticle also generated more gp350D123 protein-specific and L350 protein-specific memory B cells in splenocytes compared to L350 protein monomers. The favorable safety of the L350-ferritin nanoparticle was demonstrated by histological sectioning and H&E staining of the tissues, including lungs, heart, liver, spleen, and kidneys. No significant safety issues were identified in this histological analysis.

### 4.5. Multi-Epitope Peptide Vaccine

Multi-epitope peptide vaccines are another trend in designing EBV vaccines. Dasari et al. developed an engineered EBV poly-epitope protein, named EBVpoly, which contains 20 CD8^+^ T cell epitopes from eight different lytic and latent EBV antigens [92]. Previous peptide vaccines focused on glycoprotein gp350, which can induce both antibody and T cell responses. However, Dasari et al. aimed to induce a stronger T cell response, given that T cell-mediated immunity is critical to long-term control of latent EBV infection, which has failed to be achieved by many approaches. EBVpoly was designed by linking the CD8^+^ T cell epitopes selected to reach a broad coverage of HLA in different populations and cover the entire phase of the EBV life cycle. A lymph node-targeted amphiphile (AMP)-modified CpG DNA adjuvant, AMP-CpG, was included to enhance lymphatic immune activation. The EBV gp350 protein was also incorporated because it elicited neutralizing antibodies alongside CD8^+^ and CD4^+^ T cell responses. EBV-specific CD8^+^ and CD4^+^ T cell responses were elicited in HLA transgenic mice. The authors also compared the lymph nodes-targeted adjuvant AMP-CpG with soluble CpG. The results in HLA-B*35:01 mice showed a stronger CD8^+^ T cell response with around 11% cytokine-secreting cells (secreting one to three cytokines of IFNγ, TNFα, and IL-2, 78% of them exhibited ≥2 cytokine secretion) in the AMP-CpG adjuvant group compared to around 5% in the soluble CpG group. The AMP-CpG group generated approximately a three-fold higher gp350-specific CD4^+^ T cells than the soluble CpG, with 59% polyfunctional cells compared to 33% in the soluble CpG group. The gp350-specific humoral responses were enhanced by AMP-CpG adjuvant compared to soluble CpG, characterized by increased memory antibody-secreting cells, significantly elevated peak antibody responses at week 4 and 7, which had approximately 100-fold increased neutralizing titers. EBV-specific CD8^+^ T cells remained elevated for 29 weeks post-vaccination, with 72% polyfunctional cells secreting two or three different cytokines. In vivo experiments revealed that the adoptive transfer of EBVpoly-stimulated T cells (with or without gp350-specific antibodies) inhibited EBV lymphoma outgrowth in mice, with durable polyfunctional CD8^+^ T cell responses eradicating disseminated tumor cells, highlighting a promising cell-based therapeutic strategy.

Larijani et al. used computer-based methods to predict and design a new multi-epitope peptide vaccine against EBV [93]. They utilized different tools to screen the potential target proteins, predict the epitopes of the proteins, and predict their allergenicity, homology, immunogenicity, and toxicity. Firstly, they selected several capsids, EBNAs, envelopes, and LMPs from human gammaherpesvirus 4 (strain I) and human herpesvirus 4 (strain II). Immunogenicity was predicted to be better in two capsids and four EBNAs, and one of the LMPs of strain I. They then predicted B-cell epitopes, MHC I and MHC II epitopes, predicted their allergenicity and immunogenicity, and chose the best epitope. Finally, they created a vaccine candidate by linking sixteen epitopes from two different viral strains to each other with appropriate linkers (KK). Adjuvant Ov-ASP-1 was added to both ends of this vaccine by a proper linker (EAAAK). They then investigated the secondary structure, measured the amount of alpha-helix, and measured the stability of the candidate vaccine. This study designed a vaccine candidate for EBV peptide vaccine or, potentially, mRNA vaccine development by computer-based methods. However, the in vivo function of this vaccine should be determined by further animal experiments.

Alonso-Padilla et al. also designed an epitope-based EBV vaccine aided by computer-based methods [94]. They first generated a reference EBV proteome with variable residues masked. They used the Shannon entropy, H, to select the conserved regions. Those residue sites with H ≥ 0.5 were masked and excluded in the following epitope selection. In order to design the CD8 T cell epitopes, they began with 88 experimentally verified CD8^+^ T cell epitope sequences that were recognized during natural EBV infection, and selected only 9-residue epitopes (9 mers), which are the size of most peptides presented by MHC I molecules. After excluding those with H ≥ 0.5 sites and highly identical regions with human or human microbe proteins, 40 epitopes were retained, which were further reduced to 16 based on their functionality and antigenicity. Those epitopes were from EBNA3, BRLF1, EBNA6, EBNA1, BMRF1, and BZLF1. Two were excluded due to low population protection coverage (PPC). The authors then combined those epitopes to reach a set threshold of 95% PPC. They found that the minimal number of combinations is five, and the maximum PPC appeared in one combination of six. For the CD4^+^ T cell epitopes, they identified a total of 21 EBV-specific epitopes elicited in natural infection, which were then reduced to 10 after conservation and homology selection. For the B cell epitopes, 247 unique linear epitope sequences, ranging from 4 to 38 amino acids, were primarily selected. However, only 9 (7 from gp350 and 2 from gB) of them were not expressed intracellularly. Those two from gB were mapped onto the inner and transmembrane regions and were excluded. Flexibility and accessibility were measured in those seven epitopes from gp350, and only three of them were readily accessible for antibody recognition.

Naz et al. designed three potent peptide vaccines by integrating machine learning-based algorithms and structural biology approaches in an immunoinformatics pipeline [95]. They analyzed the EBV genome and selected nine proteins (gp42, gp350, gB, gH, gL, gM, gN, portal, and tegument proteins) with significant antigenic scores from VaxiJen. Glycoproteins, such as gp350 and gp42, were prioritized due to their high abundance on the viral envelope. The authors predicted a total of 386 linear B cell epitopes via ABCpred and validated them using different IEDB methods, of which gp350, gB, and gH contained the highest epitope density. They then analyzed the surface accessibility, hydrophilicity, flexibility, and antigenicity of these candidate regions. For T cells, MHC class-I and class-II epitopes were predicted using global allele-frequency data, followed by stringent filtering to optimize vaccine candidacy. These epitopes were further evaluated by their ethnic group coverage, antigenicity, and toxicity. The final chimeric vaccines were evaluated by population coverage, physicochemical characteristics, structural modelling, disulfide engineering, interaction with host receptors, binding patterns, discontinuous B-cell epitope prediction, molecular dynamic simulation, and immune simulation. Immune simulation using the CIMMSIM immune server showed a significant immune response after the third dose, which involved increased active B-cells and higher T cell responses.

### 4.6. mRNA Vaccine

With the approval of two rapid response mRNA vaccines for COVID-19 by the FDA, nearly a decade of clinical experience in mRNA vaccines for infectious diseases and cancer has been transformed and further developed [96,97]. Wolff et al. proposed the concept as early as 1990, but due to mRNA stability and delivery difficulties, this technology is difficult to validate [98]. The important advantage of mRNA vaccines is that they rely on the patient’s own cells for protein synthesis after entering the human body. The synthesized protein can undergo human post-translational modifications and be better delivered to the appropriate location in the cellular environment [99].

Zhao et al. developed an mRNA-based therapeutic vaccine against EBV-associated NPC. This vaccine-based epitope domain of the EBV-truncated latent proteins that bind to the TCR, including the truncated latent membrane protein 2A (Trunc-LMP2A), the truncated EBV nuclear antigen 1 (Trunc-EBNA1), and Trunc-EBNA3A, which can activate T cell anti-tumor immunity [100]. By truncating EBV antigen fragments, the antigen recognition epitopes of T cells are preserved while ensuring that these fragments lose their oncogenic functional domains. This vaccine utilizes cationic liposomes composed of diacylphosphatidylethanolamine (DOPE) and cationic lipid N-[1-(2,3-dienoxy) propyl]-N, N, N-trimethylammonium chloride (DOTMA), which can form stable nanoparticles with mRNA to achieve a spleen-targeted delivery [100,101].

In summary, gp350-based subunit vaccines are the best developed (some reaching phase I/II trials), showing protective efficacy against infectious mononucleosis but limited impact on asymptomatic infections. Viral vector-based vaccines have entered early clinical phases but face challenges in the aspects of safety and immunogenicity, while nanoparticle and mRNA vaccines remain in preclinical stages, demonstrating promising immunogenicity in animal models but are yet to be tested in humans. Multi-epitope peptide vaccines are an emerging vaccine strategy in recent years, but have not yet been applied in clinical trials. The design of multi-epitope peptide vaccines has shown broad-spectrum, high efficacy, and safety advantages, especially suitable for pathogens with frequent mutations (such as EBV). In the future, combining structural biology, computational design, and novel delivery systems is expected to break through the bottleneck of immunogenicity and epitope optimization, and promote clinical translation (Table 1).

## 5. Conclusions

The development of an effective EBV vaccine remains a critical medical need worldwide, given the virus’s global prevalence and its association with malignancies, autoimmune disorders, and other severe diseases. This review highlighted the complex biology of EBV, particularly the transmission and invasion mechanisms and the immune evasion strategies, which pose unique challenges for vaccine design. Several antigens were selected for EBV vaccine design, such as gp350, gp42, gH/gL, and latency proteins, each offering distinct advantages and limitations in eliciting protective immunity. Advances in vaccine platforms, including subunit, viral-vector, and nanoparticle-based approaches, demonstrate progress in preclinical and clinical trials. However, no candidate has achieved broad efficacy in preventing EBV infection or its associated diseases in humans.

Peptide-based vaccines represent a promising strategy for EBV vaccine development due to their ability to precisely deliver immunodominant epitopes with minimal off-target effects. Computational design tools can further enhance their efficacy by predicting conformational structures and immunogenicity, enabling rational epitope selection and optimization. Compared to natural proteins, synthetic peptides often exhibit superior immunogenicity, likely attributable to their engineered structural stability and precise antigenic focus [87,91]. A broad coverage of different antigenic epitopes might provide a wider immunization coverage towards different strains of pathogens [92]. Current peptide vaccines mainly focus on gp350 and its epitopes [102]. Despite its outstanding potency in eliciting a humoral response, whether gp350-related epitopes can elicit a satisfactory T cell response simultaneously is a concern. To address this, future designs could integrate gp350-derived B-cell epitopes with conserved T-cell epitopes from early antigens (e.g., EBNA3, BRLF1), leveraging multi-epitope platforms like lipid-conjugated peptides or nanoparticle delivery systems to enhance cross-presentation and CD8^+^ T-cell activation. However, epitope-based peptide vaccines have intrinsic shortcomings. Linear epitopes may fail to replicate conformational epitopes critical for neutralizing antibodies, and short peptides often suffer from poor immunogenicity without adjuvants or delivery carriers [87]. Advances in structural bioinformatics and chemical modifications could better preserve native-like conformations. Additionally, combining peptide vaccines with novel adjuvants or heterologous prime-boost regimens may broaden immune responses.

Four clinical trials of the gp350-based vaccine have shown success in seroconversion after vaccination. However, the preventive effect of EBV infection, especially asymptomatic infection, remains suboptimal. Combining multiple antigens and selecting novel adjuvants might improve the vaccines to reach their optimal protective effect. In summary, while hurdles persist, the convergence of innovative antigen selection, platform technologies, and adjuvant systems provides a roadmap toward an effective EBV vaccine with profound implications for public health.

## Figures and Tables

**Figure 1 viruses-17-00936-f001:**
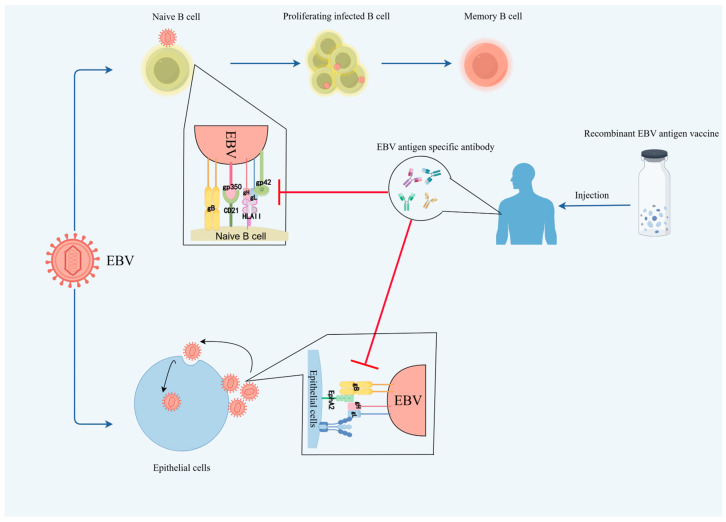
Mechanisms of EBV infection and vaccine intervention. EBV can infect two major cell types, including naive B cells and epithelial cells. B cell infection is initiated via CD21-gp350 and HLA II-gp42 interactions, driving proliferation followed by differentiation into latently infected memory B cells that serve as lifelong viral reservoirs. Epithelial cell infection is mediated by the cellular protein EphA2 that triggers membrane fusion via gH/gL-gB interaction, primarily supporting lytic replication for viral dissemination, with occasional roles in oncogenesis. To counteract those viral invasion mechanisms, recombinant EBV antigen vaccines are generated, which can stimulate the production of EBV antigen-specific antibodies in the human body. These specific antibodies can bind to the EBV, blocking its binding to host cell surface receptors, thereby preventing EBV infection of host cells and playing a role in preventing and resisting EBV infection. The schematic diagram is created with FigDraw: www.figdraw.com (accessed on 27 June 2025).

**Figure 2 viruses-17-00936-f002:**
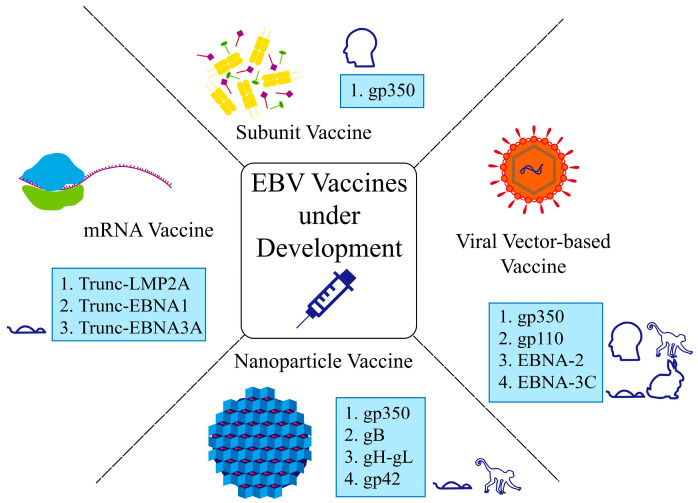
Classification of EBV vaccine candidates by platform and target antigens. Symbols (human, monkey, mouse, rabbit) adjacent to antigens indicate the experimental models used in those vaccine development studies.

**Table 1 viruses-17-00936-t001:** Summary of past and ongoing clinical studies with EBV vaccines.

Producer	Target	Vaccine Type	Phase	Ref.
GlaxoSmithKline Biologicals	gp350	subunit vaccine	II	[77,78]
Sokal et al.	gp350	subunit vaccine	II	[78]
The Cancer Research UK Formulation Unit	gp350	subunit vaccine	I	[79]
Elliott et al.	EBV-specific CD8^+^ T cells	subunit vaccine	I	[80]
Mackett and Arrand	gp350	viral vector-based vaccine	preclinical	[82]
Mackett et al.	gp350	viral vector-based vaccine	preclinical	[82]
Morgan et al.	gp350	viral vector-based vaccine	preclinical	[83]
Gu et al.	BNLF-1 MA (gp220–gp340)	viral vector-based vaccine	I	[84]
Lockey et al.	gp350, gp110,EBNA-2 and EBNA-3C	mixed viral vector-based vaccine	/	[86]
Zhang et al.	gp350 epitopes	virus-like particle-based vaccine	preclinical	[87]
Sun et al.	gB	nanoparticle vaccine	preclinical	[88]
Kanekiyo et al.	gp350	nanoparticle vaccine	preclinical	[89]
Bu et al.	gH/gL	nanoparticle vaccine	/	[45]
Zhong et al.	gH-gL, gB, and gp42	cocktail nanovaccines	preclinical	[90]
Li et al.	gp350 epitopes	nanoparticle & epitope vaccine	preclinical	[91]
Zhao et al.	Trunc-LMP2A, Trunc-EBNA1, and Trunc-EBNA3A	mRNA vaccine	preclinical	[100]
Dasari et al.	T cell epitopes	Multi-epitope peptide vaccine	preclinical	[92]
Larijani et al.	multiple epitopes	Multi-epitope peptide vaccine	In silico design	[93]
Alonso-Padilla et al.	multiple epitopes	Multi-epitope peptide vaccine	In silico design	[94]
Naz et al.	multiple epitopes	Multi-epitope peptide vaccine	In silico design	[95]

## Data Availability

All relevant materials are available from the authors.

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
