# Peer review of "Recent Progress in the Vaccine Development Against Epstein–Barr Virus"

_viruses, 2025, doi:10.3390/v17070936_

Round 1
Reviewer 1 Report
Comments and Suggestions for Authors
In the introductory section of the review, the multiple pathologies associated with Epstein-Barr virus are delineated, including the possibility of systemic autoimmunity (lines 30-32). Singh et al. (2021) Mol Immunology. The paper "Antibodies to an Epstein-Barr virus protein that cross-react with double-stranded DNA (dsDNA) have pathogenic potential" should be included in the reference section.
Figure 1- the schematic diagram crucial to understanding both B cell and epithelial cell infection by EBV is well-thought-out and presented, but the cell receptors and viral antigens are hard to read. It would benefit the reader if a darker font were used.
Author Response
1) In the introductory section of the review, the multiple pathologies associated with Epstein-Barr virus are delineated, including the possibility of systemic autoimmunity (lines 30-32). Singh et al. (2021) Mol Immunology. The paper "Antibodies to an Epstein-Barr virus protein that cross-react with double-stranded DNA (dsDNA) have pathogenic potential" should be included in the reference section.
Response: We thank the Reviewer for such constructive suggestions! We have included more citations about EBV-related autoimmune disease such as systemic lupus erythematosus (on page 2).
2) Figure 1- the schematic diagram crucial to understanding both B cell and epithelial cell infection by EBV is well-thought-out and presented, but the cell receptors and viral antigens are hard to read. It would benefit the reader if a darker font were used.
Response: We thank the Reviewer for the suggestion. In the updated Figure 1, we have appropriately increased the font size and bolded, to make the labeling more visible.

Reviewer 2 Report
Comments and Suggestions for Authors
The EBV is a significant global health concern due to its association with various diseases, including cancers. This review highlights the current state of EBV vaccine research, focusing on biological mechanisms, potential targets, and recent advancements in vaccine platforms.
Overall comment: This is a well-written review on the recent progress in therapeutics against Epstein-Barr virus (EBV). However, there are still some major and minor revisions required.
Minor edits:
Erratum line 64: EBV is one of the eight known gamma human herpesviruses,
Correction: Either EBV is one of the two known gamma human herpesviruses or one of the eight known human herpesviruses.
Erratum line 331: WT laboratory strain of vaccinia virus is a better viral vector..
Correction: WR laboratory strain of vaccinia virus is a better viral vector...
In section 3.2 line 150, please introduce the concept of tropism switching and provide a relevant citation. Also, there are several missing citations in this same review paper e.g.
line 156: Bu et al. identified two distinct fragile sites for receptor binding and B cell fusion on gp42...
line 417: Bu et al found that anti-gH/gL antibodies are the principal neutralizing components against EBV...
line 393: Kanekiyo et al. analyzed different gp350 truncation variants...
line 359: Sun et al. designed a nanoparticle (NP) vaccine
Please correct some grammar mistakes e.g. missing "," in line 27 after the text "currently".
Major edits required:
This review paper failed to highlight advances in peptide vaccines against EBV. Notable publications include https://www.nature.com/articles/s41467-023-39770-1 and https://pmc.ncbi.nlm.nih.gov/articles/PMC10043181/. And the presentation of peptides as virus-like particles (VLP) in https://pubs.acs.org/doi/abs/10.1021/acsami.5c00701.
Author Response
1)Erratum line 64: EBV is one of the eight known gamma human herpesviruses,
Correction: Either EBV is one of the two known gamma human herpesviruses or one of the eight known human herpesviruses.
Erratum line 331: WT laboratory strain of vaccinia virus is a better viral vector..
Correction: WR laboratory strain of vaccinia virus is a better viral vector...
Response: We thank the Reviewer's corrections, and have modified the manuscript accordingly.
2) In section 3.2 line 150, please introduce the concept of tropism switching and provide a relevant citation.
Response: We thank the Reviewer for the suggestion. We have added the related information in the revised manuscript (on page 4).
3) Also, there are several missing citations in this same review paper e.g.
line 156: Bu et al. identified two distinct fragile sites for receptor binding and B cell fusion on gp42...
line 417: Bu et al found that anti-gH/gL antibodies are the principal neutralizing components against EBV...
line 393: Kanekiyo et al. analyzed different gp350 truncation variants...
line 359: Sun et al. designed a nanoparticle (NP) vaccine.
Response: We thank the Reviewer's advice. In the updated manuscript, we have included these citations.
4)Please correct some grammar mistakes e.g. missing "," in line 27 after the text "currently".
Response: We thank the Reviewer’s suggestion. We carefully checked and tried the best to correct the grammar mistakes.
5)This review paper failed to highlight advances in peptide vaccines against EBV. Notable publications include https://www.nature.com/articles/s41467-023-39770-1 and https://pmc.ncbi.nlm.nih.gov/articles/PMC10043181/. And the presentation of peptides as virus-like particles (VLP) in https://pubs.acs.org/doi/abs/10.1021/acsami.5c00701.
Response: We thank the Reviewer for such constructive suggestions! In the revised manuscript, we have followed your advice by adding a new section about peptide vaccines (on page 12). And we highlight the peptide vaccine in the Conclusion part (on page 16).
